# Expression analysis of microbial rhodopsin-like genes in *Guillardia theta*

**Masae Konno**[1,2¤a¤b]*, **Yumeka Yamauchi**[1], **Keiichi Inoue**[1,2¤b], **Hideki Kandori**[1,2]*

**1** Life Science and Applied Chemistry, Graduate School of Engineering, Nagoya Institute of Technology, Nagoya, Japan, **2** OptoBioTechnology Research Center, Nagoya Institute of Technology, Nagoya, Japan

¤a Current address: PRESTO, Japan Science and Technology Agency (JST), Kawaguchi, Saitama, Japan
¤b Current address: The Institute for Solid State Physics/ISSP, The University of Tokyo, Kashiwa, Chiba, Japan
* kandori@nitech.ac.jp (HK); masaek@issp.u-tokyo.ac.jp (MK)

**Data Availability Statement:** All relevant data are within the manuscript and its Supporting Information files.

**Funding:** This work was supported by the Japan Society for the Promotion of Science (JSPS) Grant-

## Abstract

The Cryptomonad *Guillardia theta* has 42 genes encoding microbial rhodopsin-like proteins in their genomes. Light-driven ion-pump activity has been reported for some rhodopsins based on heterologous *E. coli* or mammalian cell expression systems. However, neither their physiological roles nor the expression of those genes in native cells are known. To reveal their physiological roles, we investigated the expression patterns of these genes under various growth conditions. Nitrogen (N) deficiency induced color change in exponentially growing *G. theta* cells from brown to green. The 29 rhodopsin-like genes were expressed in native cells. We found that the expression of 6 genes was induced under N depletion, while that of another 6 genes was reduced under N depletion.

## Introduction

Microbial rhodopsins are light-receiving membrane proteins that act as light-driven ion pumps, light-driven ion channels, light-driven enzymes, and photosensors [1]. The rhodopsin protein consists of seven transmembrane helices and binds an all-trans-retinal chromophore. The all-trans-retinal chromophore binds to a lysine residue conserved in the seventh transmembrane (TM) helix of all microbial rhodopsins through a protonated retinal Schiff base (SB) linkage. For decades, since light-driven $H^+$ pump bacteriorhodopsin (BR) was discovered in *Halobacterium salinarum* (formerly *H. halobium*) [2], microbial rhodopsin had been considered a unique protein possessed by a limited number of species, such as halophilic archaea. However, metagenomic analysis in the 2000s revealed that many marine prokaryotes have ion-pumping rhodopsins [3]. In 2002, a rhodopsins in the green alga *Chlamydomonas reinhardtii* was found to act as a light-gated ion channel [4]. Histidine kinase rhodopsin (HKR) was also found in *C. reinhardtii*, which contains a histidine kinase domain and response regulator domain connected to the C-terminal side of the rhodopsin domain as the first enzymatic rhodopsin [5]. The other enzymatic rhodopsin family is found in eukaryotes, and example include rhodopsin-guanylate cyclase (Rh-GC) [6] and rhodopsin-phosphodiesterase (Rho-PDE) [7]. Moreover, a new group of rhodopsins named heliorhodopsin (HeR) was also found in nature [8]. Heliorhodopsins display less than 15% sequence identity with microbial and animal

in-Aid for Scientific Research (KAKENHI), Japan (grant numbers 17H04694 to K.I. and 25104009 and 15H02391 to H.K.). There was no additional external funding received for this study. The funders had no role in the study design, data collection and analysis, decision to publish, or preparation of the manuscript.

**Competing interests:** The authors have declared that no competing interests exist.

rhodopsins. In the membrane, HeR is oriented in the opposite direction to the other rhodopsins. It is now revealed that microbial rhodopsins are widely distributed in not only bacteria but also cyanobacteria, algae and giant viruses [9–11]. Physiological roles,are related to energy production, phototaxis, regulation of gene expression, and photoautotrophy. ATP synthesis measurement in mutants of haloarchea indicates that BR and light-driven $Cl^-$ pump halorhodopsin (HR) generate proton motive force (PMF) depending on light [12, 13]. Sensory rhodopsins (SR) and channel rhodopsins act as photosensor for photomotility in *H. salinarum* [14] and *C. reinharditii* [15], respectively. *Anabaena* sensory rhodopsin (ASR) activates a soluble transducer protein (ASRT) and regulates gene transcription [16]. Proteorhodopsin (PR) contributes to phototrophy in some species of flavobacterium [17] and proteobacterium [18] in the marine environment. Recent advances in genome research have led to the discovery of many proteins that are similar to microbial rhodopsin but lack the conserved retinal-binding lysine residue (Rh-noK). There are currently 5,558 known genes of microbial rhodopsins, including HeRs, of which approximately 600 are Rh-noK. [19]. Some Rh-noK genes were tandemly arranged with PR and retinal biosynthesis genes forming a putative operon [19]. Although the molecular properties of microbial rhodopsins are studied by many approaches, studies on the physiological functions of these molecules in nature are still in progress.

Metagenomics analyses revealed that the abundance of the PR gene is negatively correlated with nitrate concentration; however, there was no significant correlation with light intensity in the north of the Sargasso Sea [20]. Nitrogen is an important source of metabolites including amino acids. Gene expression analysis of proteorhodopsin-containing flavobacteria *Dokdonia* sp. MED134 revealed that the carbon fixation pathway was shifted to that with anaplerotic $CO_2$ fixation under light conditions [21]. The effect of light was more significant in the poor-nutritional environment [21]. These results suggest that microbial rhodopsins are related to primary metabolisms processes in *Dokdonia* sp. MED134, such as carbon (C) and nitrogen (N) assimilation. N availability is limited in the marine environment; therefore, N availability could be the rate-limiting condition for primary metabolic processes, such as $CO_2$ fixation [22]. Although nitrate ions ($NO_3^-$) are a major source of N in the marine environment, anthropogenic activities result in loading a high concentration of chemically reduced forms such as ammonium ions ($NH_4^+$; [23]). To assimilate $NO_3^-$ into amino acids, it first needs to be reduced to $NH_4^+$.

Cryptophytes are unicellular algae ubiquitously found in marine and freshwater habitats [24]. Cryptophyte is considered a model taxon to study the evolution of plastids. Cryptophytes have a secondary plastid that has been acquired from other eukaryotes with primary plastids [25]. *Guillardia theta* is a cryptophyte isolated from coastal seawater [26]. Owing to the importance of the evolution of plastids, *G. theta* was the first cryptophyte whose nuclear genome was sequenced [27]. The nuclear genome of *G. theta* encodes many genes similar to microbial rhodopsins. While the molecular functions of some of these genes were characterized by heterologous expression systems [28], the molecular properties and physiological functions of many *G. theta* rhodopsins are currently not known. However, after the report of the genomic sequence of *G. theta*, new functional molecules have been reported, such as natural anion channels [29] and DTD-cation channels [30, 31]. Based on these studies, interest in the uses of these rhodopsin-like proteins in native cells has been garnered.

In this study, we investigated the expression pattern of rhodopsin-like genes in *G. theta* under various growth conditions. The N deficiency induced color change in exponentially growing *G. theta* cells from brown to green. The expression of 29 rhodopsin-like genes was observed in native cells. We revealed that the expression of 6 genes was induced under N depletion, while those of the other 6 genes were reduced under N depletion. We show that some of the rhodopsin-like genes are related to the regulation of N-assimilation by energy production in this organism.

## Results

### Phylogenetic analysis of microbial rhodopsin-like genes

Using the gene-specific NCBI reference sequences (RefSeq) database, we found 44 rhodopsin-like genes in the *G. theta* genome, although over 50 genes were suggested in a previous study [29]. The difference in the number of rhodopsin-like genes derived from the improvement of annotation in the database. Although two of them were predicted to encode hypothetical 7-transmembrane receptors (Gt_161042 and Gt_162503), our phylogenetic analysis showed that their sequences could not align with the other 42 *G. theta* rhodopsin and the representative rhodopsin sequences. Based on these results, it is considered that these two genes do not encode microbial rhodopsin. The other 42 genes were predicted to encode microbial rhodopsin proteins. Phylogenetic analysis indicated that most *G. theta* rhodopsins showed low similarity to representative rhodopsins from other species (Fig 1). There are five clades of *G. theta* rhodopsins. Clade D and E contain four cation channels and two anion channels, respectively. A lysine residue corresponding to K216 of BR is one of the important residues of microbial rhodopsins because it forms a Schiff base linkage with the retinal chromophore. The lysine residues are also conserved in most *G. theta* rhodopsins, although nine of the 42 rhodopsins do not have lysine residues (Rh-noK; S1 Fig). Clades A, B, and D contain two, five, and two Rh-noKs, respectively.

Using cDNAs derived from the extracted mRNAs, we re-analyzed the amino acid coding regions of Gt_164280 and Gt_120390. Gt_164280 was compared to the model transcript in the RefSeq database (S2 Fig). The nucleotides corresponding to the 451st to 462nd base were deleted in the model transcript and the 710th, 816th and 869th bases of re-analyzed sequence were replaced from T to C, T to C and G to A, respectively (S2A Fig). As a result, a four-amino acid insertion occurred in the amino acid sequence corresponding to TM4 in the predicted protein from the re-sequenced gene compared to that of the model transcript (S2B Fig). In re-analyzed sequence, Met234 and Ser287 of the model transcript were replaced to Thr and Gly, respectively. Gt_120390 was compared to both the model transcript and the genomic sequence (S3 Fig). The seventh exon predicted from the genomic sequence was an exact match to the mRNA sequence we determined, but the model transcript had a large deletion of 144 bases (S3A Fig). On the other hand, the mRNA we determined had a 44-base extension after the 10th exon to the intron region and then a stop codon was appeared. The 11th exon was deleted in our sequence. As a result, the transmembrane region of the predicted protein from the re-analyzed gene was consistent with that of the model transcript, but a 48-residue insertion and 14-residue amino acid substitution occurred in the C-terminal extension (S3B Fig). The C-terminal six residues were truncated in the predicted protein from re-analyzed transcript.

*Guillardia theta* also carries two heliorhodopsin-like genes (XM_005821825 and XM_005823076); however, we did not analyze the expression of these heliorhodopsins in this study.

### Effects of nitrogen availability in the growth of *G. theta*

*Guillardia theta* could grow in an artificial sea water-based medium. The color of cells turned reddish-brown to green in the late culture period (Fig 2). The color change reflected a difference in the carbon (C) or nitrogen (N) availability. Cells cultured in the C-deficient medium remained reddish-brown at the late stage of culture, whereas cells cultured in N-deficient medium turned green earlier than those cultured in N-sufficient medium. Growth of *G. theta* cells in aeration culture (12 h/12 h day/night) was monitored (Fig 3). The cellular growth in $NO_3^-$ as the sole N source was stopped by day 5 (Fig 3A, top panel), which was earlier than those under other conditions containing $NH_4^+$ (Fig 3A, middle and bottom panels). The

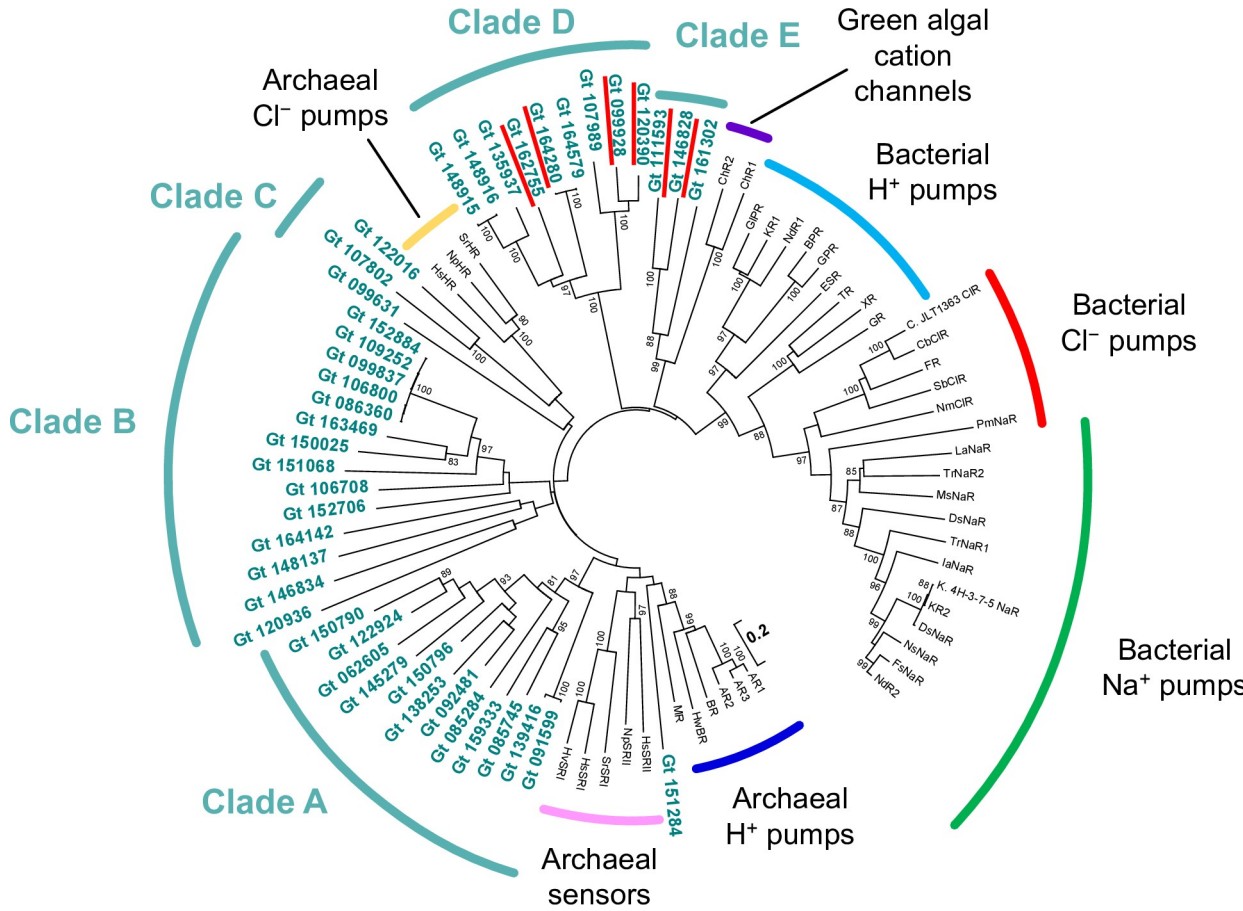

**Fig 1. The phylogenetic tree of rhodopsin-like gene expression of *Guillardia theta* (indicated in bold green) with representative microbial rhodopsins.** Percentage of replicate trees higher than 80, in which the associated taxa clustered together in the bootstrap test, is shown next to the branches. *Guillardia theta* rhodopsins whose activity has already been reported are marked by the red underlines. BR: bacteriorhodopsin from *Halobacterium salinarum*, AR1: archaerhodopsin-1 from *Halorubrum chaoviator*, AR2: archaerhodopsin-2 from *Halobacterium sp.* AUS-2, AR-3: archaerhodopsin-3 from *Halorubrum sadomense*, MR: middle rhodopsin from *Haloquadratum walsbyi*, HwBR: bacteriorhodopsin from *Haloquadratum walsbyi*, HsSRI: sensory rhodopsin I from *Halobacterium salinarum*, HvSRI: sensory rhodopsin I from *Haloarcula vallismortis* ATCC 29715, HsSRII: sensory rhodopsin II from *Halobacterium salinarum*, NpSRII: sensory rhodopsin II from *Natronomonas pharaonis*, SrSRI: sensory rhodopsin I from *Salinibacter ruber* M8, HsHR: halorhodopsin from *Halobacterium salinarum*, NpHR: halorhodopsin from *Natronomonas pharaonis*, SrHR: halorhodopsin from *Salinibacter ruber* DSM 13855, ChR1: channelrhodopsin 1 from *Chlamydomonas reinhardtii*, ChR2: channelrhodopsin 2 from *Chlamydomonas reinhardtii*, GlPR: proteorhodopsin from *Gillisia limnaea*, KR1: proteorhodopsin from *Krokinobacter eikastus*, NdR1: proteorhodopsin from *Nonlabens dokdonensis*, GPR: Proteorhodopsin from uncultured marine gamma proteobacterium, BPR: Blue-absorbing Proteorhodopsin from uncultured gamma proteobacterium, ESR: proteorhodopsin from *Exiguobacterium sibiricum*, TR: thermophilic rhodopsin from *Thermus thermophiles*, XR: xanthorhodopsin from *Salinibacter ruber*, GR: *Gloeobacter* rhodopsin from *Gloeobacter violaceus*, C. JLT1363 ClR: bacterial chloride pump rhodopsin (ClR) from *Citromicrobium* sp. JLT1363, CbClR: ClR from *C. bathyomarinum*, FR: ClR from *Fulvimarina pelagi*, SbClR: ClR from *Sphingopyxis baekryungensis*, NmClR: ClR from *Nonlabens marinus*, PmNaR: sodium pump rhodopsin (NaR) from *Phycisphaera mikurensis*, LaNaR: NaR from *Lyngbya aestuarii*, TrNaR1 and 2: *Truepera radiovictrix* NaR1 and 2, MsNaR: NaR from *Micromonospora* sp. CNB394, DsNaR: NaR from *Desulfofustis* sp. PB-SRB1, IaNaR: NaR from *Indibacter alkaliphilus*, K. 4H-3-7-5 NaR: NaR from *Krokinobacter* sp. 4H-3-7-5, KR2: NaR from *Krokinobacter eikastus*, DsNaR: NaR from *Dokdonia* sp. PRO95, NsNaR: NaR from *Nonlabens* sp. YIK-SED-11, FsNaR: NaR from *Flagellimonas* sp., NdNaR: NaR form *Nonlabens dokdonensis*.

growth rate in the logarithmic phase depended only on the $NH_4^+$ concentration (Fig 3A, middle and bottom panels). Greening was started in the late log phase to the stationary phase and was stimulated under N-deficient conditions. These results indicate that cell greening was caused by N deficiency in the cells. Furthermore, greening occurred when $NO_3^-$ was the sole N source but was not observed when $NH_4^+$ was the sole N source. These results indicate that $NH_4^+$ was preferred as the N source compared with $NO_3^-$ in *G. theta* cells.

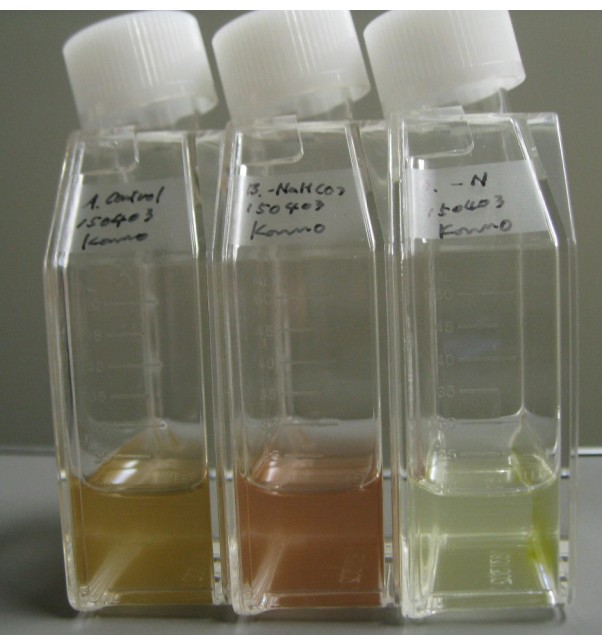

**Fig 2. Representative *Guillardia theta* cells grown in different nitrogen conditions.** Left; Normal C/N condition, Middle; C depletion, Right; N depletion.

To investigate the effect of N nutritional status on the pigment content, pigment extraction from the cells was carried out with acetone, and the change in the chlorophyll content was investigated (Fig 3B). As a result, the Chl *a* content decreased under the culture conditions where the color of the cell changed to green. In addition, the times when the Chl *a* content was less than 2 μg/$10^5$ cells and when the color change of the cell almost agreed with each other. These results suggest that chlorophyll content is one of the values that can be used as an indicator of N depletion.

## Gene expression pattern in different N availability on *G. theta*

The expression pattern of 42 microbial rhodopsin-like genes was investigated under different nitrogen conditions. The full length of each rhodopsin-like gene was amplified by reverse transcription polymerase chain reaction (RT-PCR). Twenty-nine genes could be amplified from mRNA derived from native cells (S4 Fig), indicating that these 29 genes were expressed in native cells. Among them, 25 genes could be detected using quantitative RT-PCR method. *Guillardia theta* has a predicted beta-carotene 15, 15'-monooxygenase gene (*blh*; Gt_105242), which is an enzyme that produces all-trans-retinal from beta-carotene. Based on this result, the relative expression of 25 microbial rhodopsin genes and a *blh* gene under N-deficient conditions against N-sufficient condition were quantified (Fig 4). The expression levels of nine of the 25 genes and the *blh* gene increased in N depletion (Fig 4A). Among these genes, six microbial rhodopsin genes showed > 2-fold increase in N depletion. In particular, the expression level of two genes significantly increased; Gt_120390 (*Gt*CCR1) showed 10-fold increase and Gt_111593 (*Gt*ACR1) showed a 136-fold increase in N depletion (Fig 4A, middle and right panels, respectively). The expression levels of one anion channel and two cation channels significantly increased under N depleted conditions.

The expression level of 15 genes decreased upon N depletion (Fig 4B). Among these genes, six genes showed a > 0.5-fold decrease in N depletion. The expression of the genes encodeing putative sensory rhodopsins Gt_092481 (GtR1) and Gt_085745 (GtR2), and proton pump

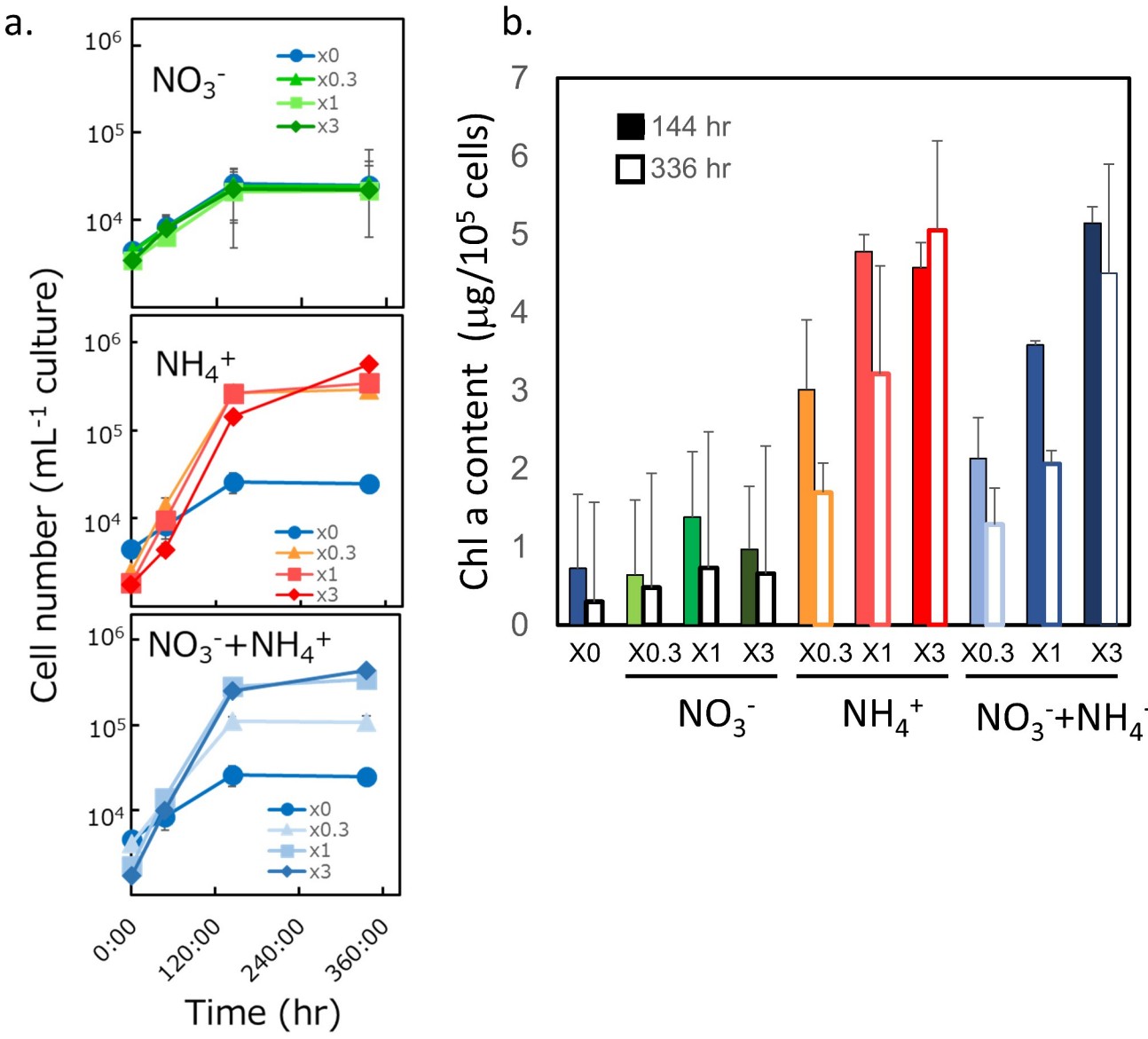

**Fig 3. Effect of nitrogen condition on growth of *Guillardia theta*.** a; The effect to growth rate, b; The effect to chlorophyll *a* content. Data are the mean value of three experimental replicates (±SD).

Gt_139416 (GtR3) were significantly suppressed in N deficiency. The gene expression of Gt_122016 (putative HKR) also tended to decrease under N-deficient conditions. Six Rh-noKs were expressed in native cells. The expression of two of six Rh-noKs, Gt_150025 and Gt_150790, was significantly suppressed in N deficiency ($< 0.5$-fold decrease). There was no difference in the gene expression of Gt_159333 regardless of nitrogen conditional.

## Discussion

### N availability of *Guillardia theta*

While $NO_3^-$ assimilation requires more reducing power than $NH_4^+$, the preferential utilization of $NH_4^+$ had inhibitory effects on the growth in aqueous culture conditions resulting from

proton imbalance [32]. Therefore, there are variations in the preference for nitrogen sources among organisms. The growth pattern of *G. theta* cells indicated that these algae prefer to use $NH_4^+$ as an N source rather than $NO_3^-$ (Fig 3). The color of *G. theta* cells turned reddish-brown to green, which was related to the nitrogen nutritional condition (Fig 2). *Rhodomonas sp*, the other species of cryptophyta, also caused the cell color change in N depletion, similar to *G, theta* [33]. In this case, phycoerythrin was preferentially degraded compared with Chl a and c under N-limiting conditions [33]. Based on this knowledge, the reason for cell color change in N depletion (Fig 2) is predicted to decrease of color pigments, such as phycobilin, carotenoid, and chlorophyll. Quantitative analysis of phycobilin and phycoerythrin is a topic of future research.

## Putative physiological functions of rhodopsin-like genes in *G. theta*

In our results, N depletion induced a decrease in chlorophyll a content (Fig 3B) and an increase in the expression of some rhodopsin-like genes (Fig 4). There was an overall negative correlation between PR gene abundance and chlorophyll a concentrations (but not light) in the surface samples and the depth profiles [20]. In the latter, there was also a negative correlation between PR genes and inorganic nutrients. The decrease in chlorophyll a content during the greening period suggested that the solar energy conversion by photosystems I and II decreased during the greening period. These results suggest that many rhodopsin-like genes are expressed under N-deficient conditions to compensate for the decrease in the utilization of solar energy. One of the physiological functions was predicted to emerge PMF for ATP synthesis [1]. Gt_139416 (GtR3) induced hyperpolarization of hippocampal neurons in response to blue light illumination and inhibited neuronal spikes [34]. Although the ion transporting activity was not characterized in detail, this study suggested that GtR3 could function as a light-dependent $H^+$ pump. The expression of GtR3 decreased with N depletion (Fig 4B), suggesting GtR3 might play a role other than the generation of PMF in N-depleted cells. The phylogenetic analysis showed that *G.theta* rhodopsin genes form five distinct clades from other representative microbial rhodopsins (Fig 1 and S1 Fig). Clade A includes GtR1 and GtR2, which have been suggested in the past as sensory rhodopsin related to phototaxis [28], but the functions of the other molecules are not known. The function of the molecules in Clade B is still unknown. This clade contains five Rh-noKs, the most numerous of all clades. The gene expression of all Rh-noK genes in clade B tended to decrease in N depletion (Fig 4). Although the function of these Rh-noK genes remains unknown, there is a possibility that these Rh-noKs has some function to modulate nitrogen homeostasis. Two of the three molecules in Clade C contain the histidine kinase domain, so they are predicted to be histidine kinase rhodopsins (HKR). The photoreaction of rhodopsin domains was reported in HKRs from *C. reinharditii* [5]. However, the histidine kinase activity of HKR as well as its physiological functions have not been revealed. Clade D has four cation channels reported to date [30, 31] and other unexplored molecules may also have cation channel activity. The Clade also contains two Rh-noKs. Clade E contains two anion channels GtACRs [29]. The channel activity of Gt_161302 was also investigated in this study; however no activity was observed [29]. GtACR1 transports $NO_3^-$ more preferentially than other anions [29], so that the increase in GtACR1 expression could enhance the transport of $NO_3^-$ in native cells to compensate for N depletion. Gt_120390 (GtCCR1) and Gt_111593 (GtACR1) showed relatively red-shifted maximum absorbance in the CCR and ACR family, respectively [29, 30]. The expression of these genes was drastically increased in N depletion (Fig 4A). The cell color change in N depletion (Fig 2) should be the cause of change in the light absorbance spectrum of *G. theta* cells compared with N sufficiency. The change in light absorbance spectrum might be the cause of the increase in GtCCR1 and GtACR1 in N depletion.

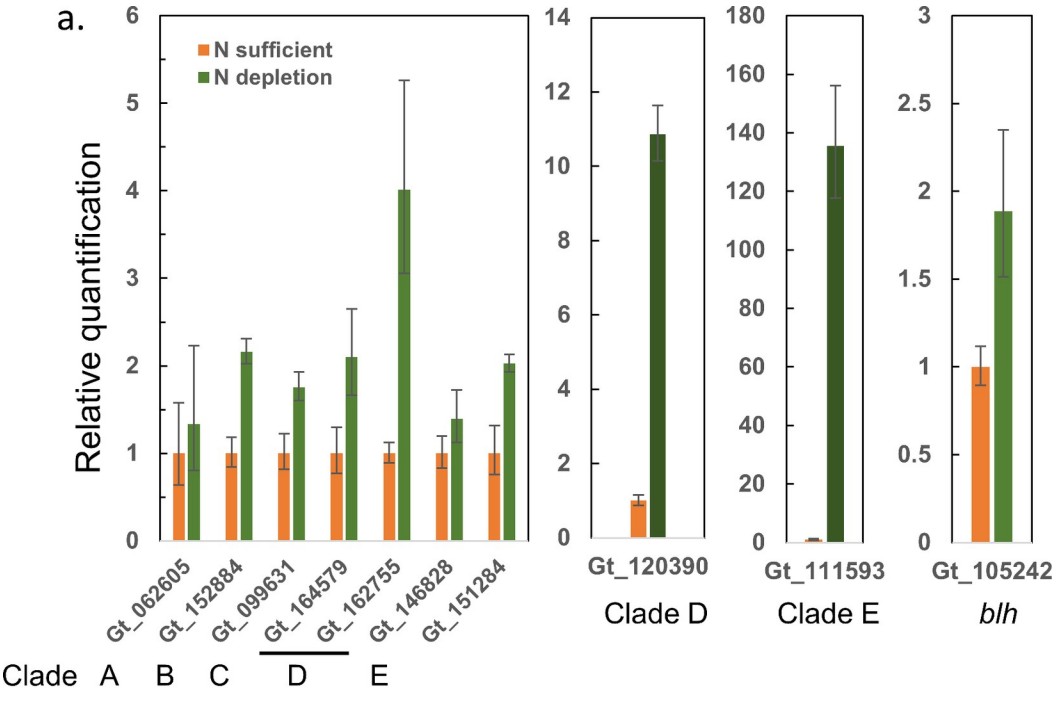

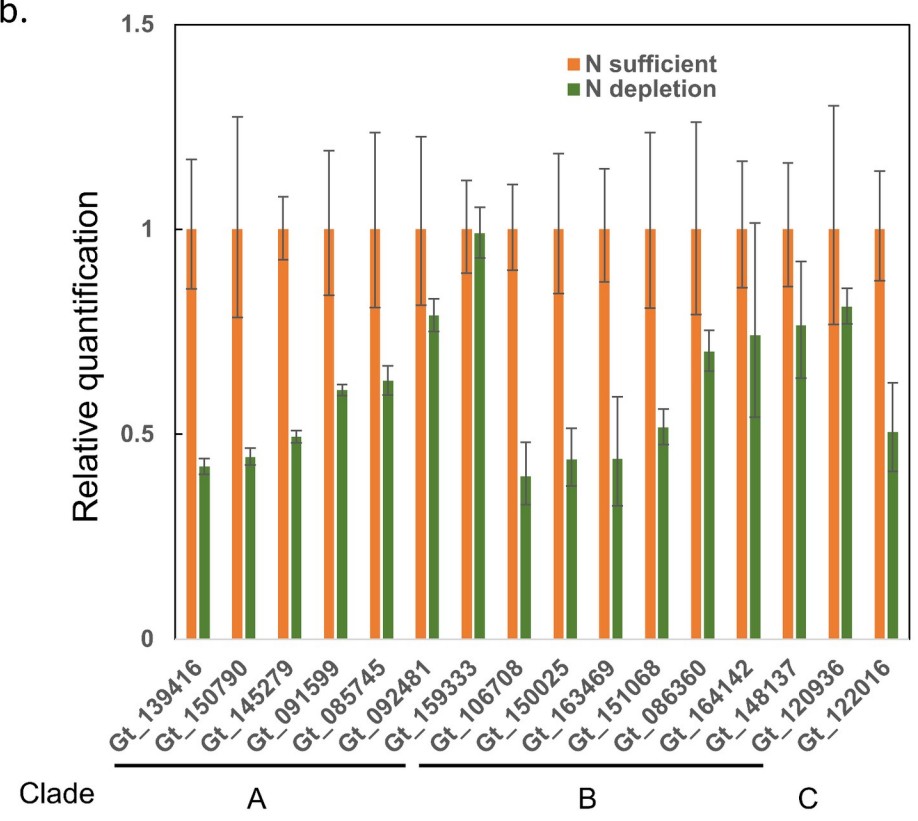

**Fig 4. Effect of nitrogen condition on rhodopsin-like gene expression of *Guillardia theta*.** a; genes of which expression levels increased in N depletion, b; genes of which expression levels decreased in N depletion. Data are the mean value of three experimental replicates. The error bar shows the lower and upper bound of the fold-change.

## Rh-noK expression

The expression of Rh-noK genes in native *G. theta* cells was first detected in this study. Unexpectedly, six Rh-noK genes were expressed in *G. theta* cells (Fig 4 and S4 Fig), indicating that these Rh-noK genes were expressed and coded as functional proteins. Rh-noK does not have a lysine residue that binds the chromophore, all-trans retinal, so it is likely to function without binding the all-trans retinal. Photoreceptor proteins such as phytochrome and cyanobacteriochrome have GAF domains that bind chromophore phytochromobilin [35]. The chromophore binds to conserved cysteine residues in the GAF domain. However, as Rh-noK has no conserved lysine residues, there are also proteins that have a GAF domain but no conserved cysteine residues [36]. Although these proteins could not bind any chromophore, they have many diverse functions, such as a sensor for sodium-ion [37] and chloride ion concentrations [36]. Many Rh-noK genes have been found in organisms such as fungi and algae [38]; therefore, they play important roles in their lives. In yeast, Rh-noKs (called ORPs) are involved in the regulation of the plasma membrane $H^+$-ATPase [39] and the maintenance of pH homeostasis [40]. It has been suggested that it has been shown to function as a chaperone [41]. The study of Rh-noK will be an interesting field for future investigation.

This study focused on the gene expression pattern of microbial rhodopsin-like genes in *G. theta* and revealed that nitrogen nutrient conditions affected to gene expression patterns. In the next step, the interest should be on the gene and/or protein expression under other circumstances, such as light conditions. The temporal and positional patterns of their expression will also be interesting to reveal the functional differentiation among these rhodopsins in native cells.

## Methods

### Cell culturing

*Guillardia theta* CCMP2712 was obtained from the Provasoli-Guillard National Center for Marine Algae and Microbiota. Cultures were grown in polyethylene culture flask in h/2 media aerated at 25°C under white light (30 mmol photons $m^{-2}$ $s^{-1}$) with light-dark cycle (12 h: 12 h). The growth rate was monitored by the optical density at 730 nm (OD 730). The value of OD 730 was correlated with the cell number which was counted with a cell counting plate (Fukaekasei, Japan). The cell number was then calculated according to the equation described below.

$$\text{Cell number (cells/mL)} = 4.1 \times 10^6 \times \text{OD}_{730} \tag{1}$$

**Measuring chlorophyll content.** Cells were collected from 1 mL culture by centrifugation. The collected cells were finally suspended in 90% acetone to obtain an extract containing all the pigments. The concentrations of chlorophyll *a* in the cells were determined by absorption using a UV/VIS spectrophotometer (Unicam UV 550, Thermo Spectronic, UK) and calculated according to the equations described below [42].

$$\text{Chl a content } (\mu g/mL) = 11.4754 \times A_{664} - 0.4574 \times A_{630} \tag{2}$$

Where $A_{664}$ and $A_{630}$ represent the absorptions at 664 and 630 nm, respectively.

### RNA extraction

Cells were grown under different nitrogen conditions (Table 1) based on artificial seawater media (h/2 media). The cells harvested by centrifugation at 3,000 × g for 15 min at 4°C. Total

**Table 1. Nitrogen concentration of culture condition.**

| Condition | $NO_3^-$ (M) | $NH_4^+$ (M) | Total N (M) |
|---|---|---|---|
| N×0 | 0 | 0 | 0 |
| $NO_3$×0.3 | $4.15×10^{-4}$ | 0 | $4.15×10^{-4}$ |
| $NO_3$×1 | $1.38×10^{-3}$ | 0 | $1.38×10^{-3}$ |
| $NO_3$×3 | $4.15×10^{-3}$ | 0 | $4.15×10^{-3}$ |
| $NH_4$×0.3 | 0 | $4.15×10^{-4}$ | $4.15×10^{-4}$ |
| $NH_4$×1 | 0 | $1.38×10^{-3}$ | $1.38×10^{-3}$ |
| $NH_4$×3 | 0 | $4.15×10^{-3}$ | $4.15×10^{-3}$ |
| $NO_3$+$NH_4$×0.3 | $2.65×10^{-4}$ | $1.50×10^{-4}$ | $4.15×10^{-4}$ |
| $NO_3$+$NH_4$×1 | $8.82×10^{-4}$ | $5.00×10^{-4}$ | $1.38×10^{-3}$ |
| $NO_3$+$NH_4$×3 | $2.65×10^{-3}$ | $1.50×10^{-3}$ | $4.15×10^{-3}$ |

cellular RNA was isolated using the RNeasy mini kit (QIAGEN, Germany) with a manufacturing protocol.

## RT-PCR to amplify full length of microbial rhodopsin genes

The RNA was reverse-transcribed using the SMARTerTM RACE cDNA Amplification Kit (Takara Bio, USA). Oligo-dT primers and random primers (N-15) were used for the first strand synthesis. The full-length microbial rhodopsin-like genes were amplified using Q5 High-Fidelity DNA Polymerase (New England Biolabs, USA) and gene-specific primers based on the mRNA sequences in the NCBI database (Table 2). Gene-specific primers contained restriction enzyme recognition sites to clone each gene to the pET21a vector.

## Reverse-transcription quantitative PCR (RT-qPCR)

The RNA was reverse-transcribed using ReverTra Ace® qPCR RT Master Mix with gDNA remover (TOYOBO, Japan). The expression levels of each microbial rhodopsin-like gene were determined using real-time PCR assay. Real-time PCR was performed on an Eco™ Real-Time PCR System (Illumina, USA) using 1 ng total RNA eq. of cDNA for each sample. THUNDER-BIRD™ SYBR® qPCR Mix (TOYOBO, Japan) was used to detect products, and 10 μM primers were used. The relative amount of cDNA in each sample was normalized using Gt_95624 (encoding an actin gene), and the melting curve was used to verify specificity. PCR was initially set at 95˚C for 60 s, followed by 42 cycles of 95˚C for 15 s and 60˚C for 60 s. The melting curve was set at 95˚C for 15 s, 55˚C for 15 s, and 95˚C for 15 s. Each gene-specific primer was based on the mRNA sequences in the NCBI database (Table 3).

## Phylogenetic analysis

The amino acid sequences used for phylogenetic analysis were those registered in the NCBI Reference Sequences (RefSeq) except for Gt120390 and Gt164280. The amino acid-coding sequences of Gt120390 and Gt164280 were re-analyzed and deposited to the public database (accession numbers are MF039475 and LC591948, respectively). The evolutionary history was inferred using the neighbor-Joining method [43]. The optimal tree with the sum of branch length = 33.09495591 is shown. The tree is drawn to scale, with branch lengths in the same units as those of the evolutionary distances used to infer the phylogenetic tree. The evolutionary distances were computed using the Poisson correction method [44] and are in the units of the number of amino acid substitutions per site. The analysis involved 85 amino acid

**Table 2. Primer sequences for amplification of full-length microbial rhodopsin genes.**

| Gene name | Accession number of mRNA | Primer sequence 5' - 3' | |
| --- | --- | --- | --- |
| | | Forward primer (5'– 3') | Reverse primer (5'– 3') |
| Gt_62605 | XM_005842420.1 | GATGCTAGCATGGCAGCGTGCGCGACG | GATCCTCGAGGACAAACTCTCTCTGTG |
| Gt_85284 | XM_005837311.1 | GATCATATGGATGTGAGTTCTGCCG | GATCCTCGAGGATGTAAGATTCAACG |
| Gt_85745 | XM_005836281.1 | GATCATATGGTCGAGGAGGGGATG | GATCCTCGAGCACGTATTCTTTTTGAG |
| Gt_86360 | XM_005834275.1 | GATCATATGGCAGCAGCGATGGGACG | GATCCTCGAGAACATATTGCTGGGAG |
| Gt_91599 | XM_005841277.1 | GATCATATGGCCTCTTCCTTCGG | GATCCTCGAGAATGGATTCATATCCTG |
| Gt_92481 | XM_005838710.1 | GATCATATGGTGCTCCAGCAGCTTGC | GACTAGCGGCCGCCACATACTCCTGGTTC |
| Gt_99631 | XM_005841908.1 | GATCATATGAGGTCGATCCAATTTTTATTG | GATCCTCGAGGTCGTTTCCGATATTCTC |
| Gt_99837 | XM_005841278.1 | GATCATATGTCTGAGGTATCGGAGGC | GATCCTCGAGGACGTACTGCTGGGAG |
| Gt_99928 | XM_005841372.1 | GATCATATGGTTGCGAGCAGCGCTG | GACTAGCGGCCGCGAAGTCCGACTCGCG |
| Gt_106708 | XM_005834183.1 | GATCATATGTCGAGCCTGACTAACGC | GATCCTCGAGGACTAGGGAAGTCTTCTC |
| Gt_106800 | XM_005834276.1 | GATCATATGCTTTTTGAGGCAGAGC | GATCCTCGAGGACGTACTGCTGGGAGGAC |
| Gt_107802 | XM_005833107.1 | GATCATATGACAGGATCGGTCCCCGG | GACTAGCGGCCGCCAACTTGCGGGTGGG |
| Gt_107989 | XM_005832876.1 | GATCATATGGTTTCTTCGAGCGCAG | GATCCTCGAGCCTGATACTTCTCTTGGC |
| Gt_109252 | XM_005831750.1 | GATCATATGGCCTTCGCTGGACCC | GATCCTCGAGAACGTACTGCTGGGAGG |
| Gt_111593 | XM_005829240.1 | GATCATATGTCAAGCATCACCTGCG | GATCCTCGAGAGCCGAATCATCATGCTC |
| Gt_120390 | XM_005820365.1 | GATGCTAGCATGGTGGAAAGCAGTGCAG | GATCCTCGAGTCCACACGCAGCAGCTTC |
| Gt_120936 | XM_005819792.1 | GATCATATGCCGGGGCTGATGTGGC | GACTAGCGGCCGCCTGCCTTGCCTTGGG |
| Gt_122016 | XM_005818716.1 | GATCATATGGAAAAGGAGCGGAGAG | GATCCTCGAGAACGCGACTTCCTTTC |
| Gt_122924 | XM_005817793.1 | GATCATATGTTCCAGATTCTCGC | GATCCTCGAGAACGAACTCCTTTGAC |
| Gt_135937 | XM_005836710.1 | GATCATATGACTGATTCAACTCCG | GATCCTCGAGCTCAGGCGTGAACATGATG |
| Gt_138253 | XM_005833449.1 | GATCATATGGCCATTGAAAGTCTGTC | GATCCTCGAGGACGTATTCCTTATCCG |
| Gt_139416 | XM_005831714.1 | GATCATATGCTCGTTGGGGAGGGCGC | GATCCTCGAGGATGGATTCGTAGCCAG |
| Gt_145279 | XM_005823958.1 | GATCATATGCCAGCGGCGGTGCAGGC | GATCCTCGAGAACATATTCACGAGTTC |
| Gt_146828 | XM_005821925.1 | GATCATATGGCAAGCCAAGTCG | GACTAGCGGCCGCGCACATGGAATGATC |
| Gt_146834 | XM_005821936.1 | GATCATATGAGCACAACTCAAAACTC | GATCCTCGAGAAAAATGAAAGTAGATTC |
| Gt_148137 | XM_005820041.1 | GATCATATGGCGAGGATGAGGGAAGG | GATCCTCGAGGAGATTGTCTTCCAGGTC |
| Gt_148915 | XM_005818945.1 | GATCATATGAGCACCTCTTCGGTAGC | GATCCTCGAGGACGTCACGTGACATC |
| Gt_148916 | XM_005818946.1 | GATCATATGAGCACCTCTTCGGTAGC | GATCCTCGAGCACATCACGCGACAT |
| Gt_150025 | XM_005841409.1 | GATGCTAGCATGTTCATTGGAGCTATCTG | GATCCTCGAGAACATACTGGTTGGTG |
| Gt_150790 | XM_005838658.1 | GATCATATGCTGGAGATGCTGAACG | GATCCTCGAGGACATATTCGCGAGACTG |
| Gt_150796 | XM_005838672.1 | GATCATATGCCGTTCGCTATGCTCGC | GACTAGCGGCCGCAACATATTCGCGAGCC |
| Gt_151068 | XM_005837691.1 | GATCATATGGCTGGAGCCGCAGGGG | GATCCTCGAGAGTCAAAGTGGCTCCGCTC |
| Gt_151284 | XM_005837266.1 | GATCATATGGTCGAGCTTACAAGTAC | GACTAGCGGCCGCTGTTCTGTAGGAATC |
| Gt_152706 | XM_005832311.1 | GATCATATGGGTCCCATCTACTAC | GATCCTCGAGTACGAACGTGCTGCTCG |
| Gt_152884 | XM_005831772.1 | GATGCTAGCATGGCTCAGTTCGCTTCCC | GATCCTCGAGAACGTACTGCTGGGAGG |
| Gt_159333 | XM_005838038.1 | GATCATATGGCGGTCCAAGATG | GATCCTCGAGTACATACTCCTTCTCCAC |
| Gt_161302 | XM_005839022.1 | GATCATATGAGCGTCGTATACGGAG | GATCCTCGAGTCTGAGGTACTCAGGGGC |
| Gt_162755 | XM_005833981.1 | GATCATATGGTTTCTGCATTGGATC | GACTAGCGGCCGCAATCGAGCGACCCGCTC |
| Gt_163469 | XM_005831796.1 | GATCATATGTGGACTGGCATCGG | GATCCTCGAGGACGTATTGCTGAGAC |
| Gt_164142 | XM_005829249.1 | GATCATATGCGCGTGAATCGGCTATG | GATCCTCGAGATCCCTCTGCTCCAGCTC |
| Gt_164280 | XM_005828899.1 | GATCATATGACGACGTCTGCCCCTTC | GATCCTCGAGAACGGCCTCGGACTCCTGC |
| Gt_164579 | XM_005827826.1 | GATCATATGGCGACGTCTGCCCCTT | GATCCTCGAGCATTCTTTCATCATCTTGC |

sequences. All ambiguous positions were removed for each sequence pair. There were a total of 409 positions in the final dataset (S1 Dataset). Evolutionary analyses were conducted in MEGA6 [45].

**Table 3. Primer sequences for RT-qPCR.**

| ID | Gene | Forward primer (5'– 3') | Reverse primer (5'– 3') |
|---|---|---|---|
| Gt_085284 | rhodopsin | AGCCATGACGGCTTGGATCG | TGCAACGCCTTGCTCAGATG |
| Gt_085745 | rhodopsin | GGCGTCCTTCTCCTATTTTGCG | CAGCCAGCAGCCCAATGTTC |
| Gt_086360 | rhodopsin | GAGCGAGACTGTGCCCCTTA | AGTGCCGAGTGAAATCTAAAGCA |
| Gt_091599 | rhodopsin | GCTTCCATCGCATACTTCTCC | GCTCACTCCAGCAACAAGACC |
| Gt_092481 | rhodopsin | TGGGAGGTTATCTGGGCACG | AAGTTGACGGCGAGAGCGTA |
| Gt_099631 | rhodopsin | GGGTTTGCGGTCCTCTACC | GTCTTGCCTGCTTCCTCTGCTTG |
| Gt_106708 | rhodopsin | GACTCAGCAGGCAAGGAACG | TTGGTCAGGTCGCAGATGG |
| Gt_111593 | rhodopsin | GCCCAATGTCACTCAAGGTGG | TCGCTCATAATACACGCTCCTG |
| Gt_120390 | rhodopsin | AACCTCAACGACCCACCAGC | AAACCATCTTCTTCGGTAAACTCCG |
| Gt_120936 | rhodopsin | TCTCCCTTCAGCATCGTCATCC | AAGCCAGCCACCAAAAGAGC |
| Gt_122016 | rhodopsin | CACTCAAGCAGGCGACACAAG | ATCAGCCAGAGGGCGAACAG |
| Gt_139416 | rhodopsin | TGGCTCATCACAACTCCCCTC | GCCCCAGTAGCAATCATCAAGAC |
| Gt_145279 | rhodopsin | ACAGTGAAGTGGTTCTGGTTCC | TCCTCCCTTGTTCTCGGCTG |
| Gt_146828 | rhodopsin | GCAAGCCAAGTCGTTTATGGAG | CTGACAACAGCCCAGACCAG |
| Gt_148137 | rhodopsin | CGTGTTCATAGTGTGCTTCTCTTC | GCTACGGCGGACAAATCATCG |
| Gt_150025 | rhodopsin | ATGCCACCGACAACTCCAAG | CGCTCTCCTCAACTCCGAAAG |
| Gt_150790 | rhodopsin | CGTCAGGAGAGCAGCGAAAA | GCAAAGTGCGAGTAGAAGAACTG |
| Gt_151068 | rhodopsin | TGCTGGGTGCCGATTTCTTG | CGATGCGGTAGATGAAGGTTGA |
| Gt_151284 | rhodopsin | CATCCTGCTGCTCACAACCC | AGAGGGACCACCCAACACAG |
| Gt_152884 | rhodopsin | TCTCACACATCCACACATCGG | GGAATAGAGCGAGGCACACTTG |
| Gt_159333 | rhodopsin | AAGTGGTTGTGGTTCTTTTTCGG | GTTGCTTGTCGGGGTTCTCC |
| Gt_162755 | rhodopsin | TGGATTGGGTTCATTGCCTTATTTG | ACATCATCTTGGTCTGGTCCTTTG |
| Gt_163469 | rhodopsin | CAGGAGGTTGGACTCATCACC | TGCTGGTTCTGGGCATCTGTG |
| Gt_164142 | rhodopsin | CTTCCATCCCAACACAGACGG | CCAAACGGTGAGACCCCAAC |
| Gt_164579 | rhodopsin | CTTCGCCCTGCTCAAGTTCCAG | CCTCGTCGTAGTTGTATCCCGC |
| Gt_105242 | blh | TGTGTGTGGCATTCGTGTCG | CTCGGTCTGCTCCAGTCTCA |
| Gt_095624 | actin | CAGAAAGGGGTGATGGTGGGA | TAGGATGGGATGCTCGTCGG |

## Supporting information

**S1 Fig. Conservation of the key amino acids for *Guillardia theta* rhodopsins compared with representative microbial rhodopsins in Fig 1.** The name of representative rhodopsins indicated in the legend of Fig 1. The color of cells is described below. Negatively charged residues: red, positively charged residues: blue, aromatic residues: grey, polar residues: light green, non-polar residues: pale yellow, methionine and cysteine: orange, histidine: dark green, glutamine and asparagine: light blue, glycine: pink, proline: black.
(TIF)

**S2 Fig. Re-analyzed sequence encoding the amino acid of Gt_164280.** a. Sequence comparison with model transcripts. b. Predicted amino acid sequence derived from re-analyzed sequence compared with that of model transcripts. MF039475: the sequence which we determined, XM_005828899.1: the model transcript of Gt_164280. Asterisks indicate that the two sequences are identical at the corresponding sites.
(TIF)

**S3 Fig. Re-analyzed sequence encoding the amino acid of Gt_120390.** a. Sequence comparison with model transcripts and genomic sequence. b. Predicted amino acid sequence derived from re-analyzed sequence compared with that of model transcripts. LC591948: the

sequence which we determined, XM_005820365.1: the model transcript of Gt_120390, NW_005464496.1: the genomic sequence of Gt_120390. Asterisks indicate that the all sequences are identical at the corresponding sites. Dots indicate that the two of three sequences are identical at the corresponding sites.
(TIF)

**S4 Fig. The full length of each rhodopsin-like gene was amplified using RT-PCR.** PCR products were detected by electrophoresis on 1% agarose gel. The lane number and ID of the expressed genes are shown in the right table. The letter and numbers are described below. M: DNA marker, 1: Gt_062605, 2: Gt_085284, 3: Gt_085745, 4: Gt_086360, 5: Gt_091599, 6: Gt_092481, 7: Gt_099631, 8: Gt_099837, 9: Gt_099928, 10: Gt_106708, 11: Gt_106800, 12: Gt_107802, 13: Gt_107989, 14: Gt_109252, 15: Gt_111593, 16: Gt_120390, 17: Gt_120936, 18: Gt_122016, 19: Gt_122924, 20: Gt_135937, 21: Gt_138253, 22: Gt_139416, 23: Gt_145279, 24: Gt_146828, 25: Gt_146834, 26: Gt_148137, 27: Gt_148915, 28: Gt_148916, 29: Gt_150025, 30: Gt_150790, 31: Gt_150796, 32: Gt_151068, 33: Gt_151284, 34: Gt_152706, 35: Gt_152884, 36: Gt_159333, 37: Gt_161302, 38: Gt_162755, 39: Gt_163469, 40: Gt_164142, 41: Gt_164280, 42: Gt_164579, 43: Gt_161042, 44: Gt_162503.
(TIF)

**S1 Dataset. The amino acid sequences using phylogenetic analysis in Fig 1.**
(FAS)

## Author Contributions

**Conceptualization:** Masae Konno, Keiichi Inoue.

**Data curation:** Masae Konno, Yumeka Yamauchi, Keiichi Inoue.

**Formal analysis:** Masae Konno, Yumeka Yamauchi, Keiichi Inoue.

**Funding acquisition:** Keiichi Inoue, Hideki Kandori.

**Investigation:** Masae Konno, Yumeka Yamauchi.

**Methodology:** Masae Konno.

**Project administration:** Masae Konno, Hideki Kandori.

**Resources:** Masae Konno, Keiichi Inoue, Hideki Kandori.

**Supervision:** Masae Konno, Hideki Kandori.

**Validation:** Masae Konno, Yumeka Yamauchi.

**Visualization:** Masae Konno, Yumeka Yamauchi, Keiichi Inoue.

**Writing – original draft:** Masae Konno.

**Writing – review & editing:** Masae Konno, Keiichi Inoue, Hideki Kandori.

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
