## [Decision Letter · Decision Letter 0]

16 Oct 2020

PONE-D-20-24671

Expression analysis of microbial rhodopsin-like genes in Guillardia theta

PLOS ONE

Dear Dr. Konno,

Thank you for submitting your manuscript to PLOS ONE. After careful consideration, we feel that it has merit but does not fully meet PLOS ONE’s publication criteria as it currently stands. Therefore, we invite you to submit a revised version of the manuscript that addresses the points raised during the review process.

Two experts in the field kindly reviewed your manuscript and offered support for the work, with both stating that the paper demonstrated scientific merit. However, a number of major issues were raised that should be addressed before your manuscript will be considered further.

Also, as stated by Reviewer 2, the paper is "very poorly written and needs to be substantially revised to make it comprehensible". As such, I strongly recommend that the revised manuscript is seen/edited by a native English speaker and/or please make use of a professional editing service before resubmission.

We look forward to receiving your revised manuscript.

Kind regards,

Wayne Iwan Lee Davies, PhD

Academic Editor

PLOS ONE

Journal Requirements:

Reviewers' comments:

Reviewer's Responses to Questions

**Comments to the Author**

1. Is the manuscript technically sound, and do the data support the conclusions?

Reviewer #1: Yes

Reviewer #2: Yes

2. Has the statistical analysis been performed appropriately and rigorously? 

Reviewer #1: Yes

Reviewer #2: Yes

3. Have the authors made all data underlying the findings in their manuscript fully available?

Reviewer #1: Yes

Reviewer #2: No

4. Is the manuscript presented in an intelligible fashion and written in standard English?

Reviewer #1: Yes

Reviewer #2: No

5. Review Comments to the Author

Reviewer #1: This is a timely report on expression of rhodopsin genes in Guillardia theta under nitrogen depletion.

Only two remarks to the authors:

1. G. theta carries some very strange heliorhodopsin-like genes. Please clearly indicate in the manuscript that those were not looked after. Or if you did measure heliorhodopsin transcripts please report it.

2. In the tree in figure 1. please clearly mark G. theta rhodopsins for which known activity was previously reported.

Reviewer #2: The manuscript by Konno et al. entitled “Expression analysis of microbial rhodopsin-like genes in Guillardia theta” reports exactly what is stated in its title. G. theta ACRs have been widely used to control neurons with light, but little is known about their function in the alga itself, and even less is known about other rhodopsin genes in this organism. The Authors have determined expression levels of G. theta rhodopsin genes under normal and nitrogen-deficient conditions and found that nitrogen depletion promoted expression of some rhodopsin genes and inhibited expression of some other ones. This manuscript provides the evidence that G. theta rhodopsin genes are at least expressed and opens a possibility to study the encoded proteins. I believe this work can be published in PLOS One, but, unfortunately, the manuscript is very poorly written and needs to be substantially revised to make it comprehensible.

1) Major issues:

Line 182: “To investigate whether a change in pigment composition in culture occurred”

I don’t understand why the Authors measured chlorophyll a content. The fact that photosynthetic pigments are degraded in aging algal cultures is very well known (e.g. [PMID: 27057086]). All cryptophytes contain phicobilins in addition to chlorophyll; in G. theta it is phycoerythrin. As the Authors correctly note, the change of red-brown to green color that they observed in aging G. theta cultures was most likely owing to predominant degradation of phycoerythrin as compared to that of chlorophyll. To test this hypothesis, they would need to measure the contents of both, phycoerythrin and chlorophyll (as it has been done in Ref. [31] in Rhodomonas) rather than chlorophyll alone. But, as Guillardia and Rhodomonas are related organisms, it is very likely that the processes of pigment degradation in them are similar. So, the Authors could simply have hypothesized that the color change they observed under nitrogen-deficient conditions in Guillardia could be explained by preferential degradation of phycoerythrin compared with chlorophyll, as measured in Rhodomonas. In any case, measurements of chlorophyll alone do not seem to produce any relevant information.

Supplementary Figure 1:

The sequences of Gt150796, Gt107989 and Gt135937 show blanks in the residue positions in the right part of the table. Does it mean that translation of the corresponding transcripts terminated before that? In any case, as determination of the actual transcript sequences is important, the Authors should show an alignment of the entire rhodopsin domains of the encoded proteins in the supplement and deposit their RNA data to a publicly available database. Also, in the text the Authors should comment on comparison of their 29 actual transcripts with the model transcripts predicted by the G. theta genome portal. As the expression constructs used in the previous studies (Refs. [27] and [28]) were based on the latter, possible errors in intron prediction might have influenced some of the results reported therein.

2) Lesser issues:

Lines 49-50: “In 2002, two rhodopsins in green algae Chlamydomonas reinhardtii were found to act as light-gated ion channels [4].”

Ref. [4] reports channel properties of only one C. reinhardtii rhodopsin, ChR1. Also, please change "algae" to "the alga".

Line 55-56: “Moreover, a new group of rhodopsin named heliorhodopsin also found in nature [8].”

Nowadays “new groups of rhodopsins” are found in nature almost every month. I believe the Authors should briefly explain here what is so new about heliorhodopsins that merits mentioning them in the introduction.

Lines 57-58: “…microbial rhodopsins are widely distributed in not only bacteria but also cyanobacteria, algae, protozoa, and giant viruses [9].”

Ref. [9] reviews only prokaryotic rhodopsins, whereas algae and protozoa are eukaryotes. Moreover, to the best of my knowledge, there are no published reports of rhodopsins in protozoa. The attribution of a rhodopsin transcript to the marine ciliate Tiarina fusus [PMID: 31320556] was erroneous; in fact, this rhodopsin was derived from a cryptophyte alga used to feed Tiarina in culture.

Line 64: “photosensor for phototaxis in H. salinarum”

Phototaxis is the ability to track the direction of light, of which H. salinarum is not capable. Please change “phototaxis” to “photomotility”. Please also change the past tense to the present tense in the entire paragraph, i.e. “Sensory rhodopsins (SR) and channel rhodopsins act as …”, not “acted”.

Line 74: “flabobacteria Doktonia sp. MED134”

This genus name is spelled Dokdonia.

Line 76-77: “These results suggest that microbial rhodopsins are related to primary metabolisms”

This conclusion can only be drawn about rhodopsins in Dokdonia, not about the entire superfamily. Furthermore, this paragraph needs to be rewritten for clarity. Start with “Nitrogen is important nutrition as a source of metabolites including amino acids” and then describe the results in Dokdonia. Delete the sentence “Algae contribute suppressing an increase in CO2 concentration in the ocean by fixing CO2 by photosynthesis” as trivial and not relevant for this paragraph.

Lines 252-253: “which have been reported in the past as sensory rhodopsin related to the phototaxis”

Please change “reported” to “suggested”. The Authors apparently mean Ref. [26], in which it has been shown that the absorption maximum of heterologously expressed and purified GtR1 lies within the broad spectral region to which G. theta motility is sensitive. This result at best justifies only a suggestion that GtR1 might contribute to phototaxis, but by no means can be considered as a “report” of its function as the receptor for phototaxis! Such conclusion would require e.g. gene knockdown studies. Also, since 2005 both CCRs and ACRs have been discovered in G. theta, and these proteins are much more likely candidates for phototaxis receptors than GtR1.

3) Some of the style and grammar problems (but there are many more throughout the manuscript – please take care of them as well):

Lines 40-42: “Microbial rhodopsins are the light-receiving membrane proteins, which act as a light-driven ion pump, a light-driven ion channel, light-driven enzyme, and photosensor [1].”

Please consistently use either plural or singular in this sentence and throughout the entire paragraph.

Line 58: “On the other hand, there are about 600 genes”

The expression “on the other hand” is unsuitable here; please change to “in addition” and add “known” after “there are”.

Lines 59-60: “As physiological roles, they are related in energy production…”

This sentence follows immediately after the sentence about Rh-noK, so the reader assumes that it describes their functions! To avoid this confusion, please move the sentence about Rh-noK to the end of this paragraph.

Lines 66-67: “Proteorhodopsin (PR) contributed with phototrophy on some species of flavobacterium”

Please change to “Proteorhodopsin (PR) mediate phototrophy in some species…”.

Line 68: “…there were many kinds of research on the molecular properties of microbial rhodopsin …”

Please change to “…molecular properties of microbial rhodopsins are studied by many approaches…”

Line 86-87: “Cryptophytes are unicellular algae living in ubiquitously from marine to freshwater environments”

Change to “Cryptophytes are unicellular algae ubiquitously found in marine and freshwater habitats”.

Lines 88-89: “…a secondary plastid that had been laterally transferred from…”

The term “lateral transfer” refers to genes, not entire organelles. Please change to “acquired”.

Line 90-91: “The reason for the importance of the evolution of plastid…”

Please change to “Owing to the importance…”

Lines 92-93: “The nuclear genome of G. theta codes many genes similar to microbial rhodopsins in its nuclear genome”.

Please delete “in its nuclear genome” at the end of this sentence.

Lines 94-95: “many of them were revealed neither their molecular functions nor physiological functions”

Please change to “Molecular properties and physiological functions of many G. theta rhodopsins are currently not known”.

Line 109: Please change “phylogenic” to “phylogenetic” here and throughout the manuscript.

Line 131: “The name of representative rhodopsins indicated as follows”.

Please change to “Abbreviations:”or delete this sentence altogether.

Line 195: Change “predicting” to “predicted”.

Line 211: “While there were 15 genes decreased the expression level in N depletion”

Please change to “The expression level of 15 genes decreased upon N depletion”.

Line 254: Please substitute the word “ref” with the corresponding reference number.

Lines 278-283: Please move these sentences to Introduction.

Lines 284-285: Please delete the words “In a similar case”. There is nothing similar between the rest of the sentence and the preceding text. The similar case is that some GAF domains have no critical cysteine, as some rhodopsin-like proteins have no critical lysine; please change the text accordingly.

6. PLOS authors have the option to publish the peer review history of their article (what does this mean?). If published, this will include your full peer review and any attached files.

Reviewer #1: No

Reviewer #2: **Yes: **Elena G. Govorunova

---

## [Author Response · Author response to Decision Letter 0]

2 Nov 2020

Response to editor's comment.

To improve the manuscript written in English, we made use of a professional editing service (Editage: www.editage.com) before resubmission.

All responses to the reviewers' comments are listed in the "Response to Reviewers" document.

I corrected the affiliation link to symbol #b.

---

## [Decision Letter · Decision Letter 1]

17 Nov 2020

PONE-D-20-24671R1

Expression analysis of microbial rhodopsin-like genes in Guillardia theta

PLOS ONE

Dear Dr. Konno,

Thank you for submitting your manuscript to PLOS ONE. After careful consideration, we feel that it has merit but does not fully meet PLOS ONE’s publication criteria as it currently stands. Therefore, we invite you to submit a revised version of the manuscript that addresses the points raised during the review process.

Your revised manuscript is much improved; however, there are a few minor changes suggested by Reviewer 2 that should be addressed before a final decision may be taken.

We look forward to receiving your revised manuscript.

Kind regards,

Wayne Iwan Lee Davies, PhD

Academic Editor

PLOS ONE

Reviewers' comments:

Reviewer's Responses to Questions

**Comments to the Author**

1. If the authors have adequately addressed your comments raised in a previous round of review and you feel that this manuscript is now acceptable for publication, you may indicate that here to bypass the “Comments to the Author” section, enter your conflict of interest statement in the “Confidential to Editor” section, and submit your "Accept" recommendation.

Reviewer #2: All comments have been addressed

2. Is the manuscript technically sound, and do the data support the conclusions?

Reviewer #2: Yes

3. Has the statistical analysis been performed appropriately and rigorously? 

Reviewer #2: Yes

4. Have the authors made all data underlying the findings in their manuscript fully available?

Reviewer #2: Yes

5. Is the manuscript presented in an intelligible fashion and written in standard English?

Reviewer #2: Yes

6. Review Comments to the Author

Reviewer #2: The Authors have much improved the manuscript during revision. They have answered all my questions and also corrected most of stylistic problems. I recommend the revised version for publication in PLOS One, although I still have a few comments, as indicated below. I am sure the Authors will agree to make these suggested changes, so I do not think there is a need to send the manuscript to me again.

Lines 50-52: “In 2002, two rhodopsins in the green alga Chlamydomonas reinharditii were found to act as light-gated ion channels [4]”.

As I have already indicated in my review of the original version, Ref. [4] reports only ONE channelrhodopsin from C. reinhardtii. The second one from this organism was described in a different paper in 2003 (PMID: 14615590). The Authors state in their response letter that they have corrected this error, but in fact they have not. Please change the plural in this sentence to singular (“a rhodopsin” and “was found”), or keep the plural and add the above-mentioned paper to the reference list.

Lines 106-107: “In G. theta, some of the rhodopsin-like genes are related to the regulation of N-assimilation by energy production”.

If this is the conclusion that the Authors draw from their present study (as I understand it is), this sentence should be moved to the next paragraph, e.g. “In this study, we investigated the expression pattern of rhodopsin-like genes in G. theta under various growth conditions. We show that some of the rhodopsin-like genes are related to the regulation of N-assimilation by energy production in this organism.” If this is a conclusion from some earlier studies, a proper reference should be given.

Lines 222-223: “Guillardia theta has a predicting beta-carotene 15,223 15'-monooxygenase gene”

Please change “predicting” to “predicted”.

7. PLOS authors have the option to publish the peer review history of their article (what does this mean?). If published, this will include your full peer review and any attached files.

Reviewer #2: No

---

## [Author Response · Author response to Decision Letter 1]

17 Nov 2020

To editor

We completely agreed the reviewer’s comments, so we reply to the comments of reviewer #2 in the 'Response to Reviewers” file.

---

## [Editor Report · Decision Letter 2]

20 Nov 2020

Expression analysis of microbial rhodopsin-like genes in Guillardia theta

PONE-D-20-24671R2

Dear Dr. Konno,

We’re pleased to inform you that your manuscript has been judged scientifically suitable for publication and will be formally accepted for publication once it meets all outstanding technical requirements.

Kind regards,

Wayne Iwan Lee Davies, PhD

Academic Editor

PLOS ONE
---

## [Editor Report · Acceptance letter]

24 Nov 2020

PONE-D-20-24671R2 

Expression analysis of microbial rhodopsin-like genes in *Guillardia theta*

Dear Dr. Konno:

I'm pleased to inform you that your manuscript has been deemed suitable for publication in PLOS ONE. Congratulations! Your manuscript is now with our production department. 

Kind regards, 

on behalf of

Dr Wayne Iwan Lee Davies 

Academic Editor

PLOS ONE